# Empirical Analysis of Scaling Vision Foundation Models for Chest X-rays

**Ahmed Al-Mahrooqi**                                    AALMAHROOQI@M42.AE
**Prateek Munjal**                                         PMUNJAL@M42.AE
**Ronnie Rajan**                                            RRAJAN@M42.AE
**Marco AF Pimentel**                                    MPIMENTEL@M42.AE
**Praveenkumar Kanithi**                                 PKANITHI@M42.AE
*M42, Abu Dhabi, UAE*

**Editors:** Accepted for publication at MIDL 2025

## Abstract

Recent advancements in multimodal transformers have shown remarkable success in computer vision and natural language tasks, yet their adaptation to the clinical world remains challenging. We introduce CXformer, a vision transformer adapted for chest X-ray analysis, through systematic investigation of architectural choices and training modifications from DINOv2. Our empirical results show that using registers in ViT training, centering the teacher model's softmax outputs, and optimizing the number of heads leads to better performance. The small version of CXformer(S) (22M parameters) achieves 83.28% mean AUROC on CheXpert test set, surpassing the baseline of 80.46% achieved with vanilla DINOv2 settings. Contrary to common assumptions, our larger model CXformer(B) with 87M parameters shows similar performance at 84% mean AUROC on CheXpert, suggesting that training optimizations matter more than model size. Furthermore compared to the current state-of-the-art RAD-DINO, our CXformer(B), with 46% reduced pretraining compute (in FLOPs) achieves an average AUROC of 87.93% (vs 87.32% by RAD-DINO) on pathology image classification task evaluated across three widely used CXR datasets i.e. CheXpert, RSNA Pneumonia, and NIH CXR8. Beyond classification, CXformer also delivers competitive, and occasionally superior, performance in semantic segmentation and radiology report generation, underscoring its versatility. CXformer base and small models can be found at `https://huggingface.co/m42-health`.

**Keywords:** Vision Foundation Models, Chest X-ray, Self Supervised Learning

## 1. Introduction

Recent advancements in large language models (LLMs) have demonstrated strong performance across a variety of natural language understanding and generation tasks (Grattafiori et al., 2024; Jiang et al., 2024; Bai et al., 2023a; DeepSeek-AI et al., 2024; Christophe et al., 2024). Typically, LLMs are pretrained using large-scale unlabeled datasets which has empirically shown to help capture general domain knowledge. However, as pretraining focuses on predicting next token, these models often require finetuning or instruction tuning to perform tasks in real world settings (Wu et al.; Luo et al., 2022; Wei et al., 2022). This paradigm has since inspired extensions to multimodal contexts, where models combine visual and textual modalities to form vision-language models (VLMs) (Liu et al., 2024; Bai et al., 2023b; Xiao et al., 2023; Agrawal et al., 2024). Like LLMs, VLMs are also trained with

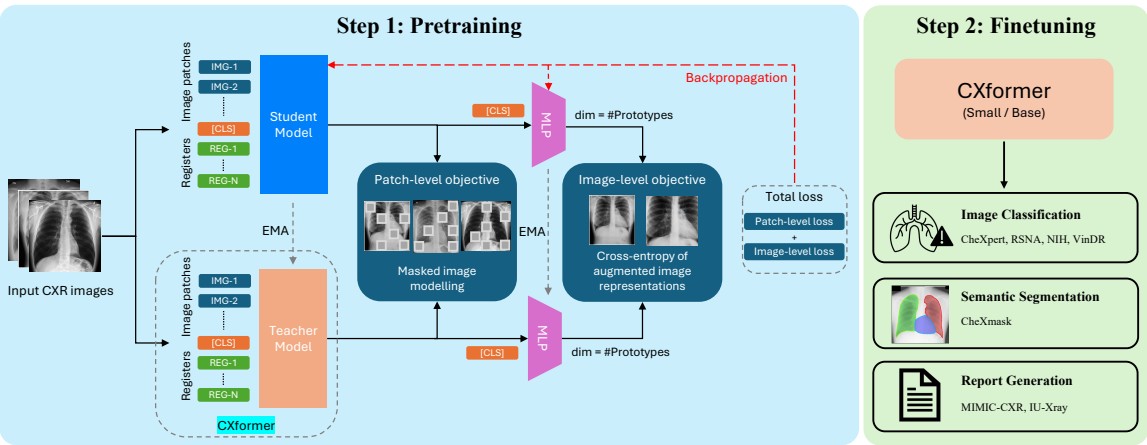

Figure 1: Overview of the CXformer development pipeline, which consists of two training stages: (1) continuous pretraining, starting from DINOv2 checkpoints, and (2) finetuning for adaptation to downstream tasks such as image classification, semantic segmentation, and report generation. During pretraining, we follow the DINOv2 training pipeline and optimize it in order to adapt to CXR images. Specifically, we incorporate the usage of registers and reduce the number of prototype heads which helps improve training efficiency without compromising downstream performance. Refer Section 3.1 for more details.

large-scale datasets such as COYO (Byeon et al., 2022), LLaVa-Instruct (Liu et al., 2023) and LLaVA-CC3M (Liu et al., 2024). Additionally, recent work has explored pretraining image encoders using contrastive learning on image-text pairs (Radford et al., 2021), with biomedical applications including (Zhang et al., 2023; Tiu et al., 2022; Khattak et al., 2024; You et al., 2023; Xu et al., 2023). However, reliance on extensive paired image-text data sets presents a significant bottleneck in medical imaging. Curating high-quality datasets at such a large scale (in the millions) is inherently difficult due to privacy concerns, data heterogeneity, and the need for corresponding expert text reports. To address this dependency, we propose developing a foundation model trained solely on large-scale Chest X-Ray (CXR) image data, which we hypothesize will enable the model to achieve robust and competitive performance across downstream tasks. Notably, methods such as DINO (Caron et al., 2021) and DINOv2 (Oquab et al., 2023) shows impressive performances (on image classification, semantic segmentation, and depth estimation tasks) on natural images, without the need of text supervision. Inspired by these methods, we employ a similar self-supervised framework to train Vision Transformers (ViTs) (Alexey, 2020) on CXR datasets, and explore its performance on downstream tasks.

Our work builds on RAD-DINO (Pérez-García et al., 2024), which utilized the DINOv2 framework for pretraining models on CXRs (Figure 1). We extend this by conducting a systematic empirical study of the key individual components within the DINOv2 training pipeline, evaluating their significance during continual pretraining on CXR datasets. Addi-

tionally, we introduce substantial modifications to the original training pipeline, simplifying it while achieving improved performance, even with smaller-sized models. We propose CXformer (Figure 1) CXR foundation models, where (S) and (B) represent ViT-S and ViT-B backbones, respectively. Our contributions are as follows: **(i)** We perform a empirical analysis on continual pretraining DINOv2 on CXRs, systematically showing the effect of individual components in training pipeline. **(ii)** We simplify the DINOv2 training pipeline by introducing key changes: incorporating registers in ViT, replacing Sinkhorn-Knopp centering with moving average centering for teacher model outputs, and reducing the number of prototype heads from $131k$ to $16k$. **(iii)** Using the simplified pipeline, CXformer(S) achieves performance comparable to SOTA RAD-DINO across tasks such as image classification, semantic segmentation, and radiology report generation, while requiring over 7 times less compute (in terms of FLOPs) and yet being approximately 3 times smaller in space complexity. **(iv)** We release both CXformer(S) and CXformer(B) models on HuggingFace along with training codebase[1] to support reproducible research in this area.

## 2. Methods

### 2.1. Models and Datasets

**Models:** Vision Transformers (Alexey, 2020) have demonstrated impressive performance (Azad et al., 2024) in medical imaging across a variety of tasks, including medical image classification, semantic segmentation, radiology report generation, multimodal VLMs (Chen et al., 2024; Li et al., 2024), and models pretrained on image-text contrastive loss (Zhang et al., 2022; Wang et al., 2022; Zhang et al., 2023). Inspired from these methods, we also conduct our empirical analysis utilizing the widely used small and base architectures of ViT, and with our proposed training pipleline (refer 2.2 for details), we term our final models as CXformer(S) and CXformer(B) models respectively.

| Dataset | Source | #Samples |
|---------|--------|----------|
| CheXpert | USA | 191,010 |
| MIMIC-CXR | USA | 237,962 |
| PadChest | Spain | 108,709 |
| NIH CXR8 | USA | 69,625 |
| BRAX | Brazil | 19,307 |
| **Total** | - | 626,613 |

Table 1: Datasets used for pretraining after filtering non-frontal views.

| Tasks | Dataset | Train | Validation | Test |
|-------|---------|-------|------------|------|
| | CheXpert | 191,010 | 202 | 500 |
| Classification | NIH CXR8 | 69,625 | 16,899 | 2,797 |
| | RSNA Pneumonia | 21,347 | 5,337 | 3,000 |
| | VinDR-CXR | 15,000 | - | 3,000 |
| Segmentation | CheXmask MIMIC-CXR | 237,923 | 1,959 | 3,403 |
| Radiology Report Generation | MIMIC-CXR | 161,923 | 1,269 | 2,461 |
| | IU-Xray | - | - | 3,309 |

Table 2: Datasets used for evaluating downstream tasks: image classification, semantic segmentation, and radiology report generation.

**Datasets:** We utilized widely used publicly available CXR datasets (refer to Table 1). For pretraining, we utilize CheXpert (Irvin et al., 2019), MIMIC-CXR (Johnson et al., 2019), PadChest (Bustos et al., 2020), NIH-CXR8 (Wang et al., 2017), and BRAX (Reis et al., 2022). For the downstream image classification task, we additionally incorporate RSNA Pneumonia (Shih et al., 2019) and VinDR-CXR (Nguyen et al., 2022). For semantic segmentation and radiology report generation, we use CheXmask (Gaggion et al., 2023) and IU-Xray (Demner-Fushman et al., 2016) dataset respectively along with the previously mentioned MIMIC-CXR dataset. These datasets provide a broad spectrum of CXRs spanning

---

1. https://github.com/m42-health/CXformer

diverse geographical regions (such as Brazil, USA and Spain), different patient demographics (e.g., age groups ranging from pediatric to elderly patients), clinical settings (inpatient and outpatient data), and a wide range of findings labeled manually by radiologists or extracted from radiology reports using NLP-based labelers (Smit et al., 2020; Irvin et al., 2019). We hypothesize that such diverse data in pretraining stage will help our models with robustness and eventually generalize better. To maintain consistency and promote reproducibility in the literature, we adhered to the official training, validation, and test splits whenever provided by the original dataset sources. Additionally, we filtered out non-frontal views, retaining only PA/AP views. The final number of datapoints retained after this filtration process are summarized in Table 1, yielding with a total of $626K$ datapoints.

| Model | Pretrain compute (exaFLOPs) | # Samples Trained |
|---|---|---|
| DINOv2 (B) | 299.52 | 1.28B |
| CheXzero | 0.03 | 1.5M |
| BiomedCLIP | 3.24 | 480M |
| RAD-DINO | 26.71 | 88.2M |
| CXformer(S) | 3.63 | 62.6M |
| CXformer(B) | 14.42 | 62.6M |

Table 3: Comparison of models based on their total pretraining compute and the number of samples used for training.

## 2.2. Experimental Setup

In traditional self-supervised learning, the base model is first pretrained using loss functions such as contrastive learning or masked language modeling. It is then finetuned for specific downstream tasks. In this section, we delve into the details of each learning stage and describe how we adapt them for CXRs in our implementation of DINOv2.

**Pretraining:** We continue pretraining (refer Figure 1) our model by initializing both the student and teacher networks from a pretrained DINOv2 checkpoint. Additionally, we incorporate registers, which introduce learnable tokens to ViTs (Darcet et al., 2023). Registers function as global memory, absorbing redundant information from low-informative patches and mitigating high-norm outliers (artifacts). Empirical studies have shown that adding registers improves performance across diverse tasks without introducing significant computational overhead. Based on this evidence, we modified the DINOv2 checkpoints to incorporate registers (Darcet et al., 2023). This register based enhancement is one of several optimizations we made to adapt DINOv2 training pipeline for CXRs. Detailed pretraining modifications are provided in Section 3.1, and additional information on pretraining can be found in Section A.1.

**Finetuning:** We evaluate our pretrained models across diverse set of tasks to understand their transferability, given the high computation requirements during pretraining. We hypothesize that a robust image foundation model should exhibit strong performance across tasks. To assess such capabilities, we perform evaluations across three tasks: **(i)** image classification, **(ii)** semantic segmentation, and **(iii)** radiology report generation. For these

experiments, we use the final teacher checkpoint as the backbone and attach a task-specific head. For example, in the CheXpert classification task, the head consists of a linear layer with five outputs. The datasets used are presented in Table 2, and their corresponding labels are shown in Table A1. We compare our models, CXformer(S) and CXformer(B), against state-of-the-art approaches, including CheXzero, BiomedCLIP (both pretrained on image-text pairs in medical domain), RAD-DINO (pretrained on CXRs), and DINOv2 (pretrained on natural images). Additionally, we provide a detailed comparison of pretraining costs in terms of compute resources and training samples in Table 3. All models, except for our lightweight ViT-S model (22M parameters), utilize the ViT-B architecture with 87M parameters. We refer the reader to A.2 for experimental setup and hyperparameters used for each task.

**Metrics:** We evaluate our pretrained models via finetuning across multiple tasks, to test our hypothesis i.e. a robust pretrained model should serve as an effective backbone for downstream tasks, including both linear probing and full finetuning. As we assess performance across three different tasks, we detail the relevant evaluation metrics in this section. Following prior works such as (Pérez-García et al., 2024), (Zhang et al., 2023), (Tiu et al., 2022), we report mean AUROC over classes (CXR findings) for the task of image classification, which evaluates the classifier's ability to differentiate between classes by measuring the area under the receiver operating characteristic curve. For semantic segmentation, we use the Dice coefficient, which quantifies the overlap between predicted and ground truth masks by comparing shared (intersection) and total pixels (union). For report generation, we assess performance using standard NLP metrics: ROUGE-L, which captures the longest common subsequence overlap between generated and reference reports; BLEU-4, which evaluates n-gram precision; $RG_{ER}$, which measures clinical coherence; and CheXbert(Smit et al., 2020) scores, which assess correctness based on classifier trained from labels extracted using Bert model instead of classic rule based CheXpert labeler, computed for both 14-label and 5-label settings from the CheXpert competition.

## 2.3. Training Details

Following Pérez-García et al. (2024), we pretrain our model for 100 epochs. Our modified training setup, adapted from the official DINOv2 implementation [2], utilizes 16 NVIDIA H100 GPUs with a global batch size of 1024. The learning rate follows a cosine decay schedule, starting at $3e-4$, with a warmup phase during the first 10% of training iterations. All input images are resized to $518 \times 518$ pixels, followed by histogram equalization. The training augmentations include resized cropping, horizontal flipping, and affine transformations such as rotation ($\pm 20°$), translation ($\pm 10\%$ in both vertical and horizontal directions), scaling (between 80% and 120%), color jittering, gaussian blurring, and solarization.

## 3. Results

In this section we evaluate our pretrained models, along with state of the art RAD-DINO, CheXzero, and BiomedCLIP (covering models trained with contrastive learning and general biomedical data beyond CXRs), as well as the pretrained DINOv2 baseline (trained

---

2. https://github.com/facebookresearch/dinov2

on natural images). All models are fully finetuned with a task-specific head and evaluated on three tasks: image classification, semantic segmentation, and radiology report generation. For each task, we report the median and 95% confidence intervals derived from 500 bootstrapped samples.

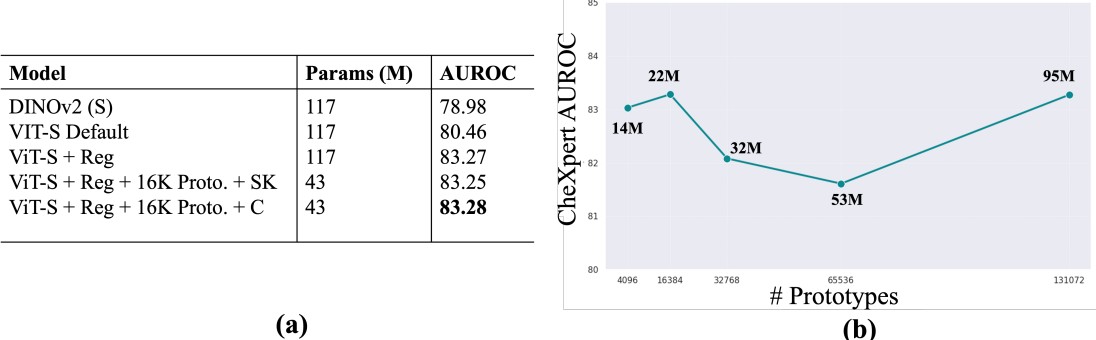

| Model | Params (M) | AUROC |
|---|---|---|
| DINOv2 (S) | 117 | 78.98 |
| VIT-S Default | 117 | 80.46 |
| ViT-S + Reg | 117 | 83.27 |
| ViT-S + Reg + 16K Proto. + SK | 43 | 83.25 |
| ViT-S + Reg + 16K Proto. + C | 43 | **83.28** |

**(a)**

**(b)**

Figure 2: **(a)** Ablations of hyper-parameters for pretraining and downstream linear probing on CheXpert. **Reg**=registers, **Proto.**=prototypes, **SK**=Sinkhorn-Knopp, **C**=centering. We indicate the number of trainable parameters for pretraining. **(b)** Effect of varying prototype number in DINO and iBOT heads on CheXpert classification. Additional parameters per prototype head are also noted.

### 3.1. Optimizing Training Pipeline

In this section, we investigate the effect of various training techniques employed in DINOv2 and evaluate their importance in adapting to CXRs. Specifically, we apply these techniques to pretrain the ViT-S model (22M parameters) using 10% of the CheXpert official training split ($\approx$ 19K datapoints). To reduce finetuning time, we adopt linear probing following (Oquab et al., 2023) evaluation protocol, systematically exploring hyperparameters–including layer selection (last 1–4), learning rates, and average pool concatenation–to mitigate potential bias, as a fixed set of hyperparameters may not be optimal across all pretrained models. We select the classifier that achieves the best performance on the validation set and report its results on the test set. By leveraging smaller models and reduced pretraining data, we address the high computational cost of pretraining, enabling faster and more efficient experimentation. The key results of this study are summarized in Figure 2(a). As a baseline, the DINOv2 (S) model achieves an AUROC of 78.98%. When continually pretrained on 10% of the CXR dataset, the ViT-S default model achieves an AUROC of 80.46%. This result is obtained using DINOv2's default settings, which include Sinkhorn-Knopp centering, 131K prototype head dimensions, and no registers. Introducing registers further improves AUROC by 2.81%, achieving 83.27%. We hypothesize that this gain comes from the register's empirical ability to reduce high-norm tokens, which are shown to cause artifacts (Darcet et al., 2023). We further explore the impact of prototype head dimensions, which in parameteric space range from 14M to 95M parameters depending on configuration. In our analysis, $16K$ as output dimensions yields best performance with results comparable to $131K$ but with almost 4x fewer parameters. Lastly, we compare Sinkhorn-Knopp centering to a simpler moving average centering of prototype scores, finding negligible differences

and favoring the latter for simplicity. The final training pipeline uses ViT with registers, 16K prototype heads, and moving average centering, along with freezing of backbone for first epoch (ref. Section A.1).

### 3.2. Image Classification Benchmark

In this section, we present our image classification results using the previously discussed optimized pipeline. To address the restriction on updates inherently imposed by linear probing, we now fully finetune the models, including the backbone and linear head. For this task, the finetuning is performed on four widely used datasets: CheXpert, RSNA Pneumonia[3], NIH CXR8[4] (from NIH Clinical Center), and VinDr-CXR Dataset, and their aggregate performances are reported in Table 4.

| Model | AUROC | | | | AUPRC |
| | CheXpert | RSNA Pneumonia | NIH CXR8 | Agg. | VinDR |
|---|---|---|---|---|---|
| DINOv2 | $78.53_{[78.25,78.53]}$ | $84.83_{[84.74,84.89]}$ | $74.85_{[74.67,74.91]}$ | 79.40 | $21.06_{[20.99,21.23]}$ |
| CheXzero | $82.48_{[82.31,82.54]}$ | $89.18_{[89.05,89.17]}$ | $77.51_{[77.37,77.57]}$ | 83.06 | $31.77_{[31.53,31.80]}$ |
| BiomedCLIP | $83.18_{[83.04,83.27]}$ | $89.54_{[89.46,89.58]}$ | $79.30_{[79.10,79.31]}$ | 84.01 | $35.85_{[35.68,35.97]}$ |
| RAD-DINO | $85.06_{[84.88,85.07]}$ | $\mathbf{92.19}_{[92.17,92.26]}$ | $84.73_{[84.53,84.71]}$ | 87.32 | $\mathbf{52.69}_{[52.49,52.76]}$ |
| CXformer(S) | $83.34_{[83.17,83.39]}$ | $91.13_{[91.03,91.13]}$ | $83.68_{[83.51,83.68]}$ | 86.05 | $46.03_{[45.96,46.24]}$ |
| CXformer(B) | $\mathbf{86.80}_{[86.67,86.85]}$ | $91.71_{[91.59,91.70]}$ | $\mathbf{85.28}_{[85.17,85.32]}$ | $\mathbf{87.93}$ | $48.02_{[47.88,48.16]}$ |

Table 4: Image classification results on CheXpert, NIH, RSNA Pneumonia, and VinDR-CXR dataset. The first four columns report AUROC, and the last column reports AUPRC. Values in brackets [ ] indicate 95% confidence intervals, computed using 500 bootstrapped samples.

Due to the data imbalance in the VinDr-CXR dataset, where normal CXRs dominate for each finding, we report the AUPRC metric, while mean AUROC is reported for the other datasets. Our exploratory data analysis revealed multiple chest X-rays from the same patients within the training split. Without patient metadata to ensure proper separation, we were unable to create an independent validation set. Instead, we trained the model on the full training set and evaluated performance using the official test set (Table 4). However, for all the other datasets we use the validation set to pick the checkpoint with best AUROC to report on the test set. This limitation also raises concerns about reproducibility in RAD-DINO (Pérez-García et al., 2024), as many of their experiments and ablations rely on the VinDr-CXR dataset rather than widely adopted datasets in the medical domain.

On an average, the DINOv2 model pretrained on natural images achieves an AUROC of 79.40%. Among models evaluated on the CheXpert and NIH datasets, CXformer(B) outperforms others, with RAD-DINO ranking second. The CXformer(B) model requires 46% less pretraining compute and 28% fewer data samples than RAD-DINO (Table 3). While our smaller model, CXformer(S), achieves a slightly lower AUROC (86.05% vs. 87.93%), it significantly reduces pretraining compute. Image-text pretrained models, such as Biomed-CLIP and CheXzero, perform worse than image-only models (CXformer(S), CXformer(B),

---

3. https://www.rsna.org/education/ai-resources-andtraining/ai-image-challenge/RSNA-Pneumonia-Detection-Challenge-2018

4. https://nihcc.app.box.com/v/ChestXray-NIHCC

and RAD-DINO), likely due to limited pretraining data, lower resolution ($224 \times 224$ for CheXzero), and larger patch sizes (32). However, BiomedCLIP shows marginally better results, likely benefiting from increased dataset diversity and compute during pretraining.

### 3.3. Segmentation and Report Generation Results

| Model | Lung | Heart | Average |
|---|---|---|---|
| DINOv2 | $91.44_{[89.87,90.60]}$ | $85.96_{[84.83,85.61]}$ | 88.70 |
| CheXzero | $84.20_{[82.90,83.64]}$ | $91.24_{[89.70,90.50]}$ | 87.72 |
| BiomedCLIP | $90.56_{[89.11,89.82]}$ | $88.38_{[87.03,87.78]}$ | 89.47 |
| RAD-DINO | $\mathbf{93.28_{[91.84,92.54]}}$ | $\mathbf{91.24_{[89.70,90.50]}}$ | **92.26** |
| CXformer(S) | $91.69_{[90.16,90.90]}$ | $89.35_{[87.62,88.49]}$ | 90.52 |
| CXformer(B) | $91.94_{[90.32,91.10]}$ | $89.94_{[87.96,88.85]}$ | 90.94 |

Table 5: Segmentation results for lung and heart anatomy using the CheXmask dataset (Gaggion et al., 2023), evaluated by the Dice score. Values in brackets [ ] indicate 95% confidence intervals, computed using 500 bootstrapped samples

In this section, we present segmentation results on the CheXmask dataset for lung and heart regions in CXRs, evaluated using Dice score. RAD-DINO achieves the highest overall performance of 92.3. Our CXformer models perform competitively, with CXformer(B) achieving an average Dice score of 90.94 and CXformer(S) scoring 90.52. These results highlight our model's strong segmentation capabilities while maintaining parameter space efficiency. BiomedCLIP achieves a solid average Dice score of 89.47, consistent performance across lungs (90.56) and heart (88.38). DINOv2 shows good lung segmentation (91.44) but underperforms for the heart (85.96). Surprisingly, CheXzero achieves the highest heart segmentation score (91.24) but struggles with lungs (84.20), leading to an average score of 87.72. Thus, CLIP-based models (CheXzero, BiomedCLIP) trained on image-text pairs demonstrate inferior performance compared to other models which aligns with the findings of (Oquab et al., 2023), which may highlight the challenges in learning localized features. Overall, the CXformer models demonstrate robust segmentation performance comparable to state-of-the-art methods, further validating their effectiveness in semantic segmentation tasks. We provide qualitative analysis in Section A.3.

Finally, we discuss the radiology report generation results, as shown in Table 6. For MIMIC-CXR, both CXformer variants deliver competitive results across datasets, with CXformer(B) showing a balance between lexical metrics and clinical metrics. Notably, the small variant, CXformer(S), outperforms all other models across all metrics except for CheXbert-F1 score, where it achieves the second-highest performance. Despite this, it requires $3\times$ fewer parameters, $7.4\times$ less pretraining compute, and 30% fewer pretraining samples. CLIP-based models (CheXzero, BiomedCLIP), despite their multimodal pretraining on paired image-text datasets, do not consistently outperform image-only pretraining models on clinically relevant metrics. The IU-Xray dataset, which was not used during training, serves as an external, out-of-domain test set, originating from an outpatient facility. Interestingly, while the performance gap between CheXzero and image-based models is

| (a) MIMIC-CXR | | | | | | |
|---|---|---|---|---|---|---|
| **Model** | **ROUGE-L** | **BLEU-4** | **RG$_{ER}$** | **CheXbert F1-14** | **CheXbert F1-5** | **Average** |
| DINOv2 | $24.24_{[24.21,24.25]}$ | $8.51_{[8.49,8.52]}$ | $21.43_{[21.42,21.46]}$ | $28.62_{[28.59,28.69]}$ | $42.09_{[42.00,42.15]}$ | 24.98 |
| CheXzero | $23.36_{[23.34,23.38]}$ | $7.95_{[7.93,7.96]}$ | $20.95_{[20.93,20.98]}$ | $29.06_{[29.04,29.13]}$ | $44.22_{[44.13,44.27]}$ | 25.11 |
| BiomedCLIP | $23.35_{[23.33,23.36]}$ | $7.71_{[7.70,7.73]}$ | $20.47_{[20.45,20.49]}$ | $28.77_{[28.73,28.83]}$ | $42.84_{[42.73,42.88]}$ | 24.63 |
| RAD-DINO | $24.91_{[24.89,24.92]}$ | $8.82_{[8.82,8.85]}$ | $22.92_{[22.91,22.96]}$ | $\mathbf{35.40_{[35.39,35.52]}}$ | $\mathbf{47.54_{[47.46,47.61]}}$ | **27.92** |
| CXformer(S) | $\mathbf{25.25_{[25.23,25.27]}}$ | $\mathbf{9.11_{[9.09,9.12]}}$ | $\mathbf{23.06_{[23.04,23.08]}}$ | $33.85_{[33.83,33.94]}$ | $46.28_{[46.14,46.28]}$ | 27.51 |
| CXformer(B) | $24.93_{[24.90,24.94]}$ | $9.03_{[9.01,9.05]}$ | $22.94_{[22.93,22.98]}$ | $33.45_{[33.39,33.50]}$ | $45.45_{[45.36,45.49]}$ | 27.16 |

| (b) IU-Xray | | | | | | |
|---|---|---|---|---|---|---|
| **Model** | **ROUGE-L** | **BLEU-4** | **RG$_{ER}$** | **CheXbert F1-14** | **CheXbert F1-5** | **Average** |
| DINOv2 | $26.85_{[26.83,26.86]}$ | $8.63_{[8.61,8.64]}$ | $26.37_{[26.36,26.39]}$ | $17.49_{[17.43,17.58]}$ | $22.26_{[22.16,22.49]}$ | 20.32 |
| CheXzero | $26.97_{[26.95,26.98]}$ | $8.71_{[8.70,8.73]}$ | $\mathbf{27.73_{[27.71,27.75]}}$ | $20.55_{[20.45,20.63]}$ | $29.93_{[29.69,30.01]}$ | 22.78 |
| BiomedCLIP | $26.65_{[26.64,26.67]}$ | $8.60_{[8.60,8.62]}$ | $27.11_{[27.10,27.13]}$ | $17.57_{[17.43,17.62]}$ | $27.21_{[26.85,27.25]}$ | 21.43 |
| RAD-DINO | $\mathbf{27.18_{[27.17,27.19]}}$ | $9.42_{[9.41,9.43]}$ | $26.98_{[26.97,27.00]}$ | $27.19_{[27.08,27.31]}$ | $\mathbf{33.16_{[33.05,33.43]}}$ | **24.79** |
| CXformer(S) | $26.47_{[26.47,26.59]}$ | $9.26_{[9.25,9.27]}$ | $26.26_{[26.23,26.27]}$ | $23.55_{[23.49,23.60]}$ | $29.76_{[29.60,29.97]}$ | 23.06 |
| CXformer(B) | $26.85_{[26.84,26.87]}$ | $\mathbf{9.54_{[9.53,9.55]}}$ | $27.02_{[26.99,27.03]}$ | $25.92_{[25.84,26.06]}$ | $31.62_{[31.46,31.84]}$ | 24.19 |

Table 6: Performance of CXR report generation evaluated on **(a)** MIMIC-CXR and **(b)** IU-Xray datasets across multiple metrics. Values in brackets [ ] indicate 95% confidence intervals, computed using 500 bootstrapped samples

narrowed in this setting, but still underperforms compared to our models and RAD-DINO. A possible explanation is the prevalence of studies with no findings ($\sim 40\%$ vs. $\sim 20\%$ in MIMIC-CXR), along with frequent findings such as cardiomegaly, hyperinflated lungs, and a tortuous aorta being among the ten most common findings (Demner-Fushman et al., 2016), which do not require fine-grained localized features. We further show and discuss qualitative samples in A.4.

## 4. Conclusion

In this work, we introduce CXformer, a specialized adaptation of DINOv2 for chest X-rays (CXRs), optimized through systematic modifications to the standard training pipeline of DINOv2. Key enhancements include integrating registers in ViTs to reduce artifacts, centering teacher outputs using a moving average, and reducing prototype heads to 16K for improved efficiency and implicit regularization. These optimizations enable more efficient training and superior downstream performance. Notably, CXformer(S) matches RAD-DINO while requiring approximately $7\times$ fewer FLOPs and $3\times$ less memory, and CXformer(B) surpasses RAD-DINO in average AUROC across CheXpert, NIH CXR8, and MIMIC dataset with a 46% FLOP reduction. Our ablation studies confirm that these improvements stem from training optimizations rather than increased model size. Additionally, we find that image-only self-supervised pretraining outperforms image-text contrastive methods (e.g., BiomedCLIP, CheXzero) by generalizing effectively without the explicit multimodal training required by image-text models. We note that while our optimizations are shown for DINOv2 adapting CXRs, their effectiveness in other medical imaging modalities, like CT/MRI imaging, remains uncertain and is an interesting direction for future research. By releasing CXformer checkpoints, we aim to foster reproducibility and drive advancements in efficient models in healthcare domain.

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

## Appendix A.

### A.1. Self-supervised pretraining

**Model**  For pretraining, we adopt the self-supervised learning framework of DINOv2 (Oquab et al., 2023). This approach leverages a teacher-student architecture, where both networks share the same architecture, but the teacher network is updated as the exponential moving average (EMA) of the student's weights. The pretraining process optimizes two complementary objectives: an image-level objective and a patch-level objective. The image-level objective computes a cross-entropy loss between features extracted by the teacher and student networks from different crops of the same image. These features are projections of the classification token, [CLS], processed through a learnable MLP head, followed by softmax and a centering operation. This objective ensures that the representations learned by the model are invariant to different augmentations of the same image. On the other hand, the patch-level objective focuses on learning localized features by masking random patches in the input image fed to the student network, while the teacher sees the full image. The masked patches are processed by the student's learnable MLP head, while the unmasked patches are processed by the teacher's MLP head, with a cross-entropy loss applied to the resulting tokens. This encourages the student network to learn fine-grained, localized information from the image, complementing the global information captured by the image-level objective.

### A.2. Downstream finetuning

| Tasks | Dataset & Label |
|---|---|
| **Classification** | **CheXpert**: Atelectasis, Cardiomegaly, Consolidation, Edema, Pleural Effusion |
| | **NIH CXR8**: Atelectasis, Cardiomegaly, Infiltration, Pleural Effusion, Mass, Nodule, Pneumonia, Pneumothorax |
| | **RSNA Pneumonia**: Pneumonia |
| | **VinDR-CXR**: Aortic enlargement, Cardiomegaly, Lung Opacity, Pleural Effusion, Pleural thickening, Pulmonary fibrosis, Tuberculosis |
| **Segmentation** | **CheXmask MIMIC-CXR**: Lungs & Heart |
| **Radiology Report Generation** | **MIMIC-CXR**: Findings Section |
| | **IU-Xray**: Findings Section |

Table A1: Overview of labels used for each dataset.

**Image Classification**  We initialize our classification experiments using the pretrained encoder backbone, appending a single classification layer on top of the vision backbone. To extract meaningful features for classification, we utilize the outputs from the last four blocks of the Vision Transformer (ViT). Specifically, we concatenate the [CLS] token embeddings

from these blocks with the average-pooled representation of the image token embeddings from the final block. This combined feature vector serves as the input to the classification head. For training, we employ a batch size of 128 over two NVIDIA H100 GPUs, setting the learning rate for the linear classifier to $5e-5$ and a smaller learning rate of $5e-7$ for the backbone to prevent overfitting and preserve the pretrained representations. Preprocessing of input images involves histogram equalization, followed by resizing the shorter edge to 518 pixels and applying a center crop of size $518 \times 518$. To enhance generalization, we apply a range of training augmentations, including horizontal flips, random affine transformations, color jittering, and Gaussian blur. The final model used for evaluation is selected based on its best performance on the validation set. For all datasets, we utilize the area under the receiver operating characteristic curve (AUROC) as the primary evaluation metric. In the case of multi-label classification tasks, we report the macro-average AUROC, which considers the performance across all labels equally. A special case is with VinDr (Nguyen et al., 2022), where we report the macro AUPRC, since there exists a significant class imbalance, and to be able to compare to similar works (Pérez-García et al., 2024), which report this metric.

**Semantic Segmentation** For the segmentation task, we append a decoder head to the pretrained backbone, implemented as a linear layer following the approach in DINOv2 (Oquab et al., 2023). This linear layer is trained to predict class logits from the patch tokens, generating a low-resolution logit map. The logit map is subsequently upsampled to a resolution of $518 \times 518$ to produce the final segmentation map. We evaluate segmentation performance using the Dice score across two classes: lungs and heart. Input images are preprocessed by histogram equalization, resizing to $518 \times 518$ pixels (resizing the shorter edge to 518 pixels, followed by center cropping), and augmented with affine transformations, color jitter, and elastic transformations. Training is performed with a batch size of 128 across 2 NVIDIA H100 GPUs, and a learning rate of $5e-3$.

**Image to Radiology Report Generation** For the radiology report generation task, we train on the MIMIC-CXR dataset (Johnson et al., 2019) (and additionally evaluate on IU-Xray dataset (Demner-Fushman et al., 2016)), which pairs chest X-ray images with their corresponding radiology reports. From the raw reports, we extract the findings section[5] and filter the data as follows: we discard samples without a findings section, retain only frontal-view images (AP and PA views), and exclude samples where the findings section contains fewer than 100 characters. After preprocessing, the dataset comprises of 161,927 samples for training, 1,269 samples for validation, and 2,461 samples for testing. For IU-Xray dataset, we end up with 3,309 samples after the filtering process. For training an image-to-report generation model, we adopt the LLaVA-1.5 framework (Liu et al., 2024) for visual instruction tuning, closely following the methodology and hyperparameters in (Hyland et al., 2023). Fine-tuning involves loading our pretrained vision backbone and appending a trainable two-layer MLP projection layer to align visual tokens with textual tokens. The projected visual tokens are combined with language instructions in the format: `<image>` `Provide a description of the findings in the radiology image`. This input is fed into a pretrained language decoder model to generate the findings section corresponding to the image. We use `Vicuna-7B (v1.5)` (Chiang et al., 2023) as the decoder language

---

5. https://github.com/MIT-LCP/mimic-cxr/tree/master

model. During training, the image encoder is kept frozen, while the projection layer and the language decoder are trainable. Training is conducted in $bfloat16$ precision with a batch size of 128 across 8 NVIDIA H100 GPUs. We employ a cosine learning rate scheduler with a peak learning rate of $2e - 5$ and a warmup phase spanning 3% of the total iterations. The standard cross-entropy loss for next-token prediction is used as the training criterion, consistent with language modeling tasks. After training, generations are produced with a temperature of 0.2 and a maximum of 150 new tokens. For evaluation, we report lexical and radiology-specific metrics, namely: BLEU-4 score (Papineni et al., 2002), ROUGE-L score (Lin, 2004), RadGraph-F1 score (Delbrouck et al., 2022), and finally CheXbert-based F1 score (Smit et al., 2020). We use the `HuggingFace Evaluate`[6] library for BLEU and ROUGE scores, and open-source implementations for CheXbert-F1[7] and RadGraph-F1[8].

### A.3. Anatomy Segmentation Qualitative Results

Figure A1: Segmentation samples of different models. Lungs and heart segmentation are highlighted in blue and green, respectively. The ground truth segmentation is outlined in red. Dice scores are reported above each segmentation.

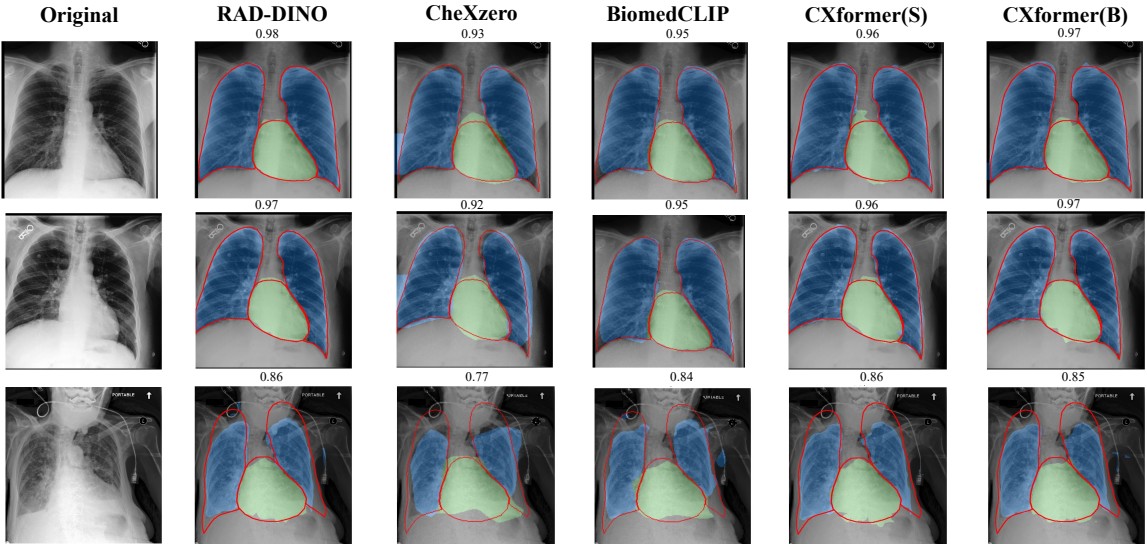

In this section, we present some qualitative examples or anatomy segmentation in Figure A.3. The first row shows an example where RAD-DINO achieves the highest average Dice score, followed by CXformer(B) and CXformer(S). Our CXformer(B) models shows better segmentation and coverage of the heart, while the smaller variant slightly overestimates the heart segmentation outside of the original region. CheXzero has slightly less precise segmentation, particularly around the lower lung and diaphragm, while Biomed-CLIP is slightly better than CheXzero, but still falls behind RAD-DINO and CXformer

---

6. https://huggingface.co/docs/evaluate

7. https://pypi.org/project/f1chexbert

8. https://pypi.org/project/radgraph

models. This same behavior is seen in the second row, where CheXzero overestimates the lung regions beyond the original object. In the last row, we present an example where the original groung-truth segmentation overestimates the lung boundaries. This discrepancy may stem from the use of a deep learning algorithm to generate the ground truth segmentation in the source dataset (Gaggion et al., 2023), potentially leading to an inaccurate assessment of model performance. In this case, RAD-DINO and our CXformer models accurately capture the lung region without overestimating the upper boundaries while also providing better heart coverage. In contrast, the CheXzero model struggles to identify the lung region and overestimates the heart beyond its actual position. Although BiomedCLIP achieves a higher Dice score, its segmentation quality remains suboptimal, particularly as the model appears to struggle in the presence of support devices.

## A.4. Radiology Report Generation Qualitative Results

Figure A2: Comparison of a report generation study, with green highlights marking correct identifications, and red highlights marking incorrect findings or hallucinations.

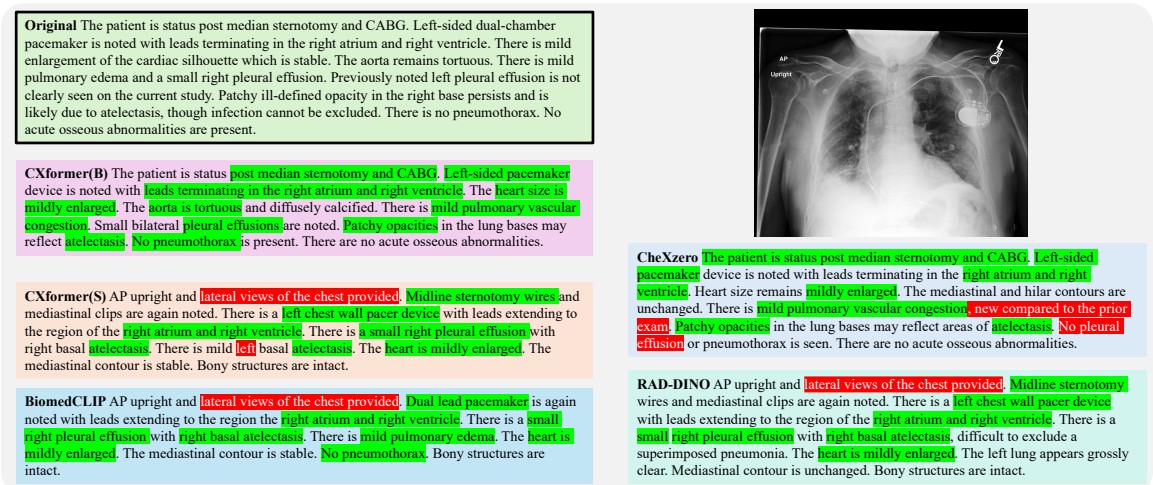

In this section, we present qualitative results for the radiology report generation task on the MIMIC-CXR (Johnson et al., 2019) test set. As shown in Figure A.4, our CXformer(B) model accurately identifies the pacemaker and its location, mild cardiomegaly, a tortuous aorta, pleural effusions, and patchy opacities (which may indicate atelectasis), as well as the absence of pneumothorax. While the original report explicitly mentions 'pulmonary edema,' the generated finding of 'mild pulmonary vascular congestion' refers to the same underlying condition. Our CXformer(S) model also correctly identifies these findings; however, it hallucinates the presence of later views (which are not provided) and incorrectly attributes atelectasis to the left side instead of the right, as stated in the original report. We note similar hallucinations of lateral views in BiomedCLIP, CheXzero and RAD-DINO models. Additionally, CheXzero incorrectly rules out pleural effusion. With the exception of our CXformer(B) model, all models fail to indicate the tortous aorta.

Figure A3: Comparison of a report generation study, with green highlights marking correct identifications, and red highlights marking incorrect findings or hallucinations.

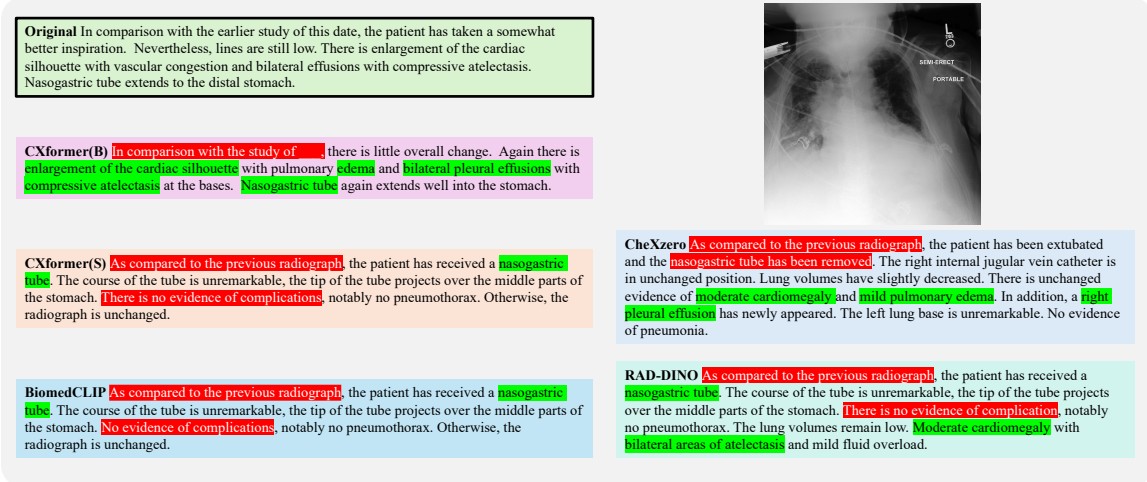

In the next example (Figure A.4), we observe that all models hallucinate the presence of a previous study. This likely stems from the original dataset, where patients in ICU settings undergoing portable X-rays often have multiple follow-up scans. Our CXformer(B) model correctly identifies 1)cardiomegaly, 2)edema, 3)bilateral pleural effusion, 4)compressive atelectasis and the 5)nasogastric tube. On the other hand, the CXformer(S) and BiomedCLIP only identifies one finding correctly (nasogastric tube), CheXzero only identifies 3 out of the five findings (cardiomegaly, edema and effusion), and lastly, RAD-DINO only identifies 3 different findings (nasogastric tube, cardiomegaly, atelactasis).

In Figure A.4, we present an unremarkable study with no findings and compare the outputs of both CXformer models. The generated reports closely match the ground truth, with identical phrasing in both generations. This consistency likely arises from the prevalence of template reports for normal scans, which describe the absence of commonly reported findings.

In Figure A.4, the ground truth report identifies a small pneumothorax, pleural effusion on the right, compressive atelectasis, and other findings with a high level of detail. The CXformer models, however, display varying degrees of hallucination, likely stemming from training data references to prior studies. CXformer(S) correctly identifies the presence of pleural effusion and notes that the central catheter remains in place, but it introduces hallucinated findings such as pulmonary edema, cardiac silhouette enlargement, and references to an earlier study, which are absent from the ground truth. On the other hand, CXformer(B) misses the pleural effusion and central catheter findings but accurately mentions the removal of the nasogastric tube, consistent with the ground truth.

In Figure A.4, the ground truth report describes moderate pulmonary edema, a small-to-moderate left pleural effusion, a probable small right pleural effusion, compressive atelectasis at the left base, and moderate cardiomegaly. Among the model generations, CXformer(S) demonstrates better coverage of the findings compared to CXformer(B). CXformer(S) cor-

Figure A4: Comparison of CXformer model generations for a normal study with no findings, showing consistent phrasing and alignment with the ground truth report.

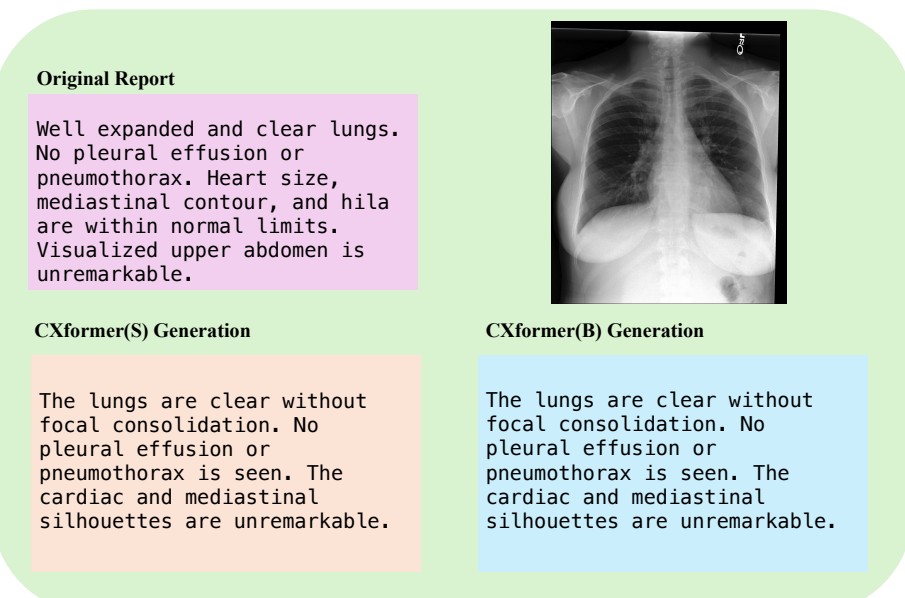

Figure A5: Example of model hallucination to previous studies, as well as some omission of findings.

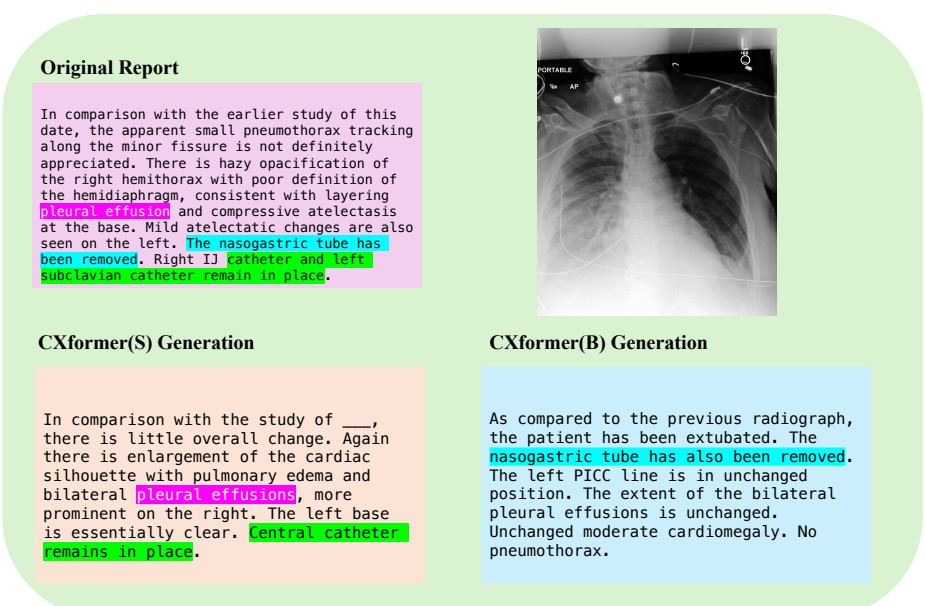

rectly identifies the moderate left pleural effusion, moderate cardiomegaly, and absence of pneumothorax, though it overestimates the pleural effusion size and understates the pul-

monary edema as mild. In contrast, CXformer(B) only highlights the enlarged heart, but omitting the pleural effusion and pulmonary edema. However, CXformer(B) uniquely identifies the portable nature of the imaging technique, which is not explicitly mentioned in the ground truth but can be inferred from the radiograph. Similarly, the same behavior is seen in Figure A.4, where CXformer(S) has better coverage of findings, whilst CXformer(B) model exhibits some omission of findings.

Figure A6: Comparison of CXformer model generations: CXformer(S) provides better coverage of findings, while CXformer(B) highlights the portable view but misses key pathologies.

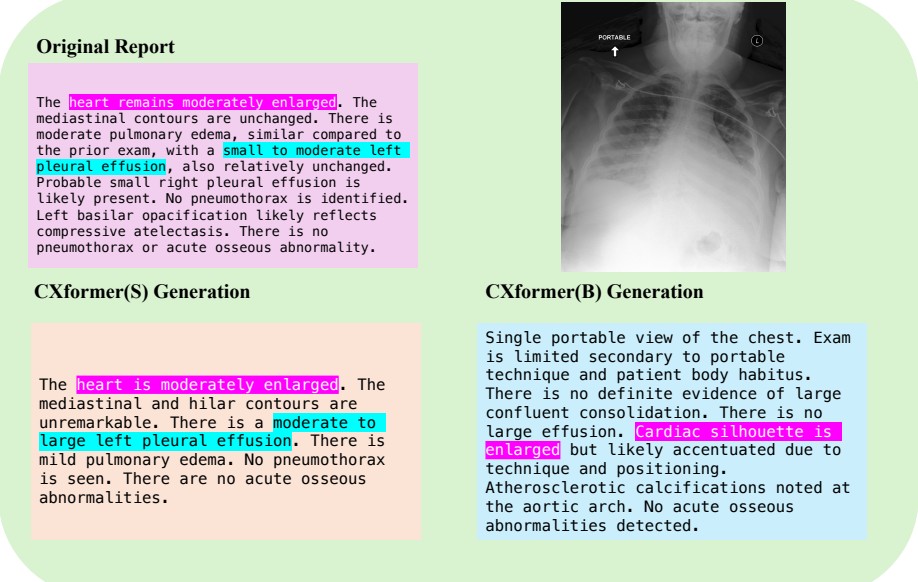

Figure A7: Comparison of CXformer model generations: CXformer(B) omits the calcification of aorta findings, while CXformer(S) correctly covers it.

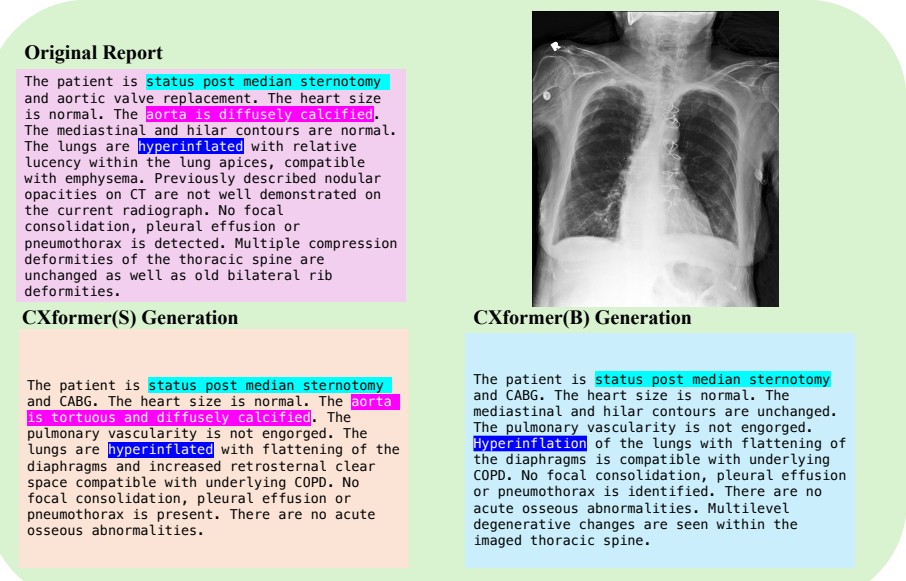

**Original Report**

The patient is status post median sternotomy and aortic valve replacement. The heart size is normal. The aorta is diffusely calcified. The mediastinal and hilar contours are normal. The lungs are hyperinflated with relative lucency within the lung apices, compatible with emphysema. Previously described nodular opacities on CT are not well demonstrated on the current radiograph. No focal consolidation, pleural effusion or pneumothorax is detected. Multiple compression deformities of the thoracic spine are unchanged as well as old bilateral rib deformities.

**CXformer(S) Generation**

The patient is status post median sternotomy and CABG. The heart size is normal. The aorta is tortuous and diffusely calcified. The pulmonary vascularity is not engorged. The lungs are hyperinflated with flattening of the diaphragms and increased retrosternal clear space compatible with underlying COPD. No focal consolidation, pleural effusion or pneumothorax is present. There are no acute osseous abnormalities.

**CXformer(B) Generation**

The patient is status post median sternotomy and CABG. The heart size is normal. The mediastinal and hilar contours are unchanged. The pulmonary vascularity is not engorged. Hyperinflation of the lungs with flattening of the diaphragms is compatible with underlying COPD. No focal consolidation, pleural effusion or pneumothorax is identified. There are no acute osseous abnormalities. Multilevel degenerative changes are seen within the imaged thoracic spine.

