# OpenReview forum: "Empirical Analysis of Scaling Vision Foundation Models for Chest X-rays"
_MIDL.io/2025/Conference — MIDL 2025 Poster_

### Official Review · Reviewer_Srhg · 2025-02-13

**Confidence:** 4
**Preliminary Rating:** 2
**Final Rating:** 3

**Summary:**

This paper presents a study of pretraining DINO-V2 on CXR dataset by utilizing several optimizing techniques to reduce the training compute, including registers in ViT training, centering the teacher model’s softmax outputs, and optimizing the number of heads. The pretrained models are validated on several downstream tasks, such as classification, semantic segmentation and radiology report generation, and shows robust and competitive performance.

**Strengths:**

1. Pretraining DINO-v2 for CXRs is an interesting topic, showing if an extra pretraining before transferring to downstream tasks is necessary.
2. The authors introduced several techniques during the pretraining stage to reduce the compute is useful.
3. The validation on downstream tasks are comprehensive, thus convincing.

**Weaknesses:**

1. The pretraining process is a bit confusing, can you provide the architecture of the model, involving all modules, such as register, and organize them in a figure with more details?
2. The models compared should be finetuned for whole network, not just the task-specific head, as the proposed model has been pretrained on CXR data, it would be unfair for comparison otherwise . And what is the fine-tuning strategy for each model?

3. the sentence said ‘we apply these techniques to pretrain the ViT-S model (22M parameters) using 10% of the CheXpert official training split (≈ 19K datapoints)’, what if using more data?

4. in table 4, besides AUROC, please also provide ACC or other accuracy metrics for more information.

**Detailed Comments:**

1. the links to the datasets used in thos work should be provided.
2. can you explain why filtered out non-fraontal views in pretraining?

3. some typing errors? such as (1) ‘a linear layer with five outputs Sare presented in Table 2, ‘; (2) ‘We refer the reader to A.2 for experimental setup and hyperparameters used for each task. ch’; (3) ‘The key results of this study are summarized in Figure 3(a).’, where is Figure 3(a)? , and so on. please polish language again.

4. the sentence said ‘Specifically, we apply these techniques to pretrain the ViT-S model (22M parameters)’, however, in Figure 2 (a) , it shows 250M, which is the right one?

5. what do the subscriptions in [] of the values in table 4 represent? please clarify that.

**Justification Of The Final Rating:**

I think the authors addressed most of my questions.

What I'm still concerned about is the fine-tuning strategy, particularly the learning rate selection, when compared to other methods for a fair comparison. As the authors said that for classification they also fine-tuned the encoders for all methods, but just simply 'setting the learning rate for the linear classifier to 5e−5 and a smaller learning rate of 5e−7 for the backbone', which I would say not suitable, even though they say they followed previous work, because the scenario is different. Regarding the computation, the proposed method also requires pretraining.

Based on the rebuttal, I would raise my rating from weak reject to borderline.

**Justification Of The Preliminary Rating:**

As mentioned above, the work about pretraing DINO-v2 for CXRs is interesting, but the description of the method is a bit unclear, providing a figure for this can be a good point. And more importantly, the experiments in finetuning phase for downstream tasks might be less convincing, as the settings such as learning rate for the encoders of all compared models are not given, and in Appendix, using lr=5e-7 might be a good choice.

**Questions To Address In The Rebuttal:**

Please tackle all the points mentioned above by either revising  descriptions  or clarifying the arguments.

---

> ### Author Response · Authors · 2025-03-07
> **Rebuttal response to Reviewer Srhg**
>
> - **W1** “The pretraining process is a bit confusing........figure with more details”?\
> We thank the reviewer for bringing this up, and we have revised the manuscript with more details. We also request the reviewer to see Figure 1 in the revised manuscript which contains more details.
> \
> &nbsp;
> - **W2** “The models compared should be finetuned for whole network ...... And what is the fine-tuning strategy for each model”?\
> We thank the reviewer for raising this point. We perform full finetuning for image classification, while for other tasks, we train only the task-specific head. This aligns with prior work like Rad-DINO and DinoV2(trained for natural images). As our goal is to test whether image foundation models are effective, we keep encoders frozen for fair comparison. Linear probing is used in ablations to save computational resources utilized to pretrain and finetune large models. Except for standard DinoV2, all other models include biomedical data, including CXRs. We have updated the manuscript to clarify these points and appreciate the reviewer’s feedback. We also request the reviewer to refer to Section A.2 in the supplementary material for technical details for each downstream tasks. Additionally, to enhance reproducibility and clarity in implementation, we will release the codebase on GitHub upon publication.
> \
> &nbsp;
> - **W3** “the sentence said ‘we apply these........(≈ 19K datapoints)’, what if using more data”?\
> We intentionally started with a smaller subset to first optimize the training pipeline for adapting DinoV2 to CXRs. Given the high cost of pretraining and fine-tuning, we prioritized systematic ablations before scaling up. In our final experiments, we used the full dataset for image classification and empirically demonstrated that our optimized pipeline allows ViT-S to perform competitively, narrowing the gap between ViT-B and ViT-S, as seen with Scan42(B) and Scan42(S). We have clarified this in the manuscript and appreciate the reviewer’s thoughtful question.
> \
> &nbsp;
> - **W4** “In table 4, besides AUROC.... metrics for more information.”?\
> We appreciate the reviewer’s suggestion and understand the request for additional metrics. In our evaluation, we follow the standard practices established by seminal works like RAD-DINO and DINOv2. Specifically, we report AUROC and AUPRC for image classification, Dice Score for semantic segmentation, and Rouge-L, BLEU-4, RGER, and CheXbert-F1-14/5 for report generation. Given the class imbalance in most medical imaging datasets, we believe accuracy may not fully capture model performance and hence is not a widely adopted metric in the literature. However, we value the reviewer’s input and are open to further discussion and potential amendments to the manuscript if it would improve clarity.
> \
> &nbsp;
> - **DC1** “the links to the datasets .... be provided.”?\
> We thank the reviewer for pointing this out. We request the reviewer to see section 2.1 (Datasets) of the revised manuscript where we have updated the links to each dataset.
> \
> &nbsp;
> - **DC2** ”can you explain why filtered out non-fraontal views in pretraining”?\
> We filtered out non-frontal views during pretraining because the majority of scans in our dataset are AP/PA, and data from non-frontal views could introduce noise during training. This approach is also aligned with similar works like Rad-DINO and CheXzero. However, we acknowledge that exploring the inclusion of other views is an interesting direction, and we address it as a future work in our revised manuscript. We appreciate the reviewer’s insightful suggestion.
> \
> &nbsp;
> - **DC3** ”some typing errors? ....... refer the reader to A.2 ....... please polish language again”?\
> We apologize for this confusion,  and we request the reviewer to see the revised manuscript where we have fixed all these mistakes.
>  \
> &nbsp;
> - **DC4** “the sentence said ‘Specifically...to pretrain the ViT-S...... , it shows 250M, which is the right one”?\
> We apologize for the confusion. The sentence mentioning ViT-S (22M parameters) refers specifically to the backbone used for all our ablation studies. However, in Figure 2(a), the reported 250M parameters represent the total parameter count, which includes not only the student and teacher model backbones but also the parameters from the DINO head and iBOT head which are used to compute the overall DINO loss used during pretraining. We request the reviewer to see Figure 1 in the revised manuscript.
> \
> &nbsp;
> - **DC5** "what do the subscriptions in [] ....please clarify that."?\
> We apologize for the confusion. The values in [ ] represent 95% confidence intervals, calculated using 500 bootstrapped samples. We have clarified this in the revised manuscript.

---

> > ### Comment · Reviewer_Srhg · 2025-03-13
> >
> > Although, except for the standard DinoV2, all other models include biomedical data, such as CXR, for a fair comparison they should also be fine-tuned with the encoder, as they are pretrained with a larger dataset, thus have domain shift, while the proposed method is further pretrained with CXR only. Moreover, to fine-tuning these models, it would be fair to select the best learning rate using the validation dataset, not just simply 'setting the learning rate for the linear classifier to 5e−5 and a smaller learning rate of 5e−7 for the backbone', because learning rate matters a lot in the final performance.

---

> > > ### Author Response · Authors · 2025-03-13
> > > **Rebuttal discussion with Reviewer Srhg**
> > >
> > > We appreciate the reviewer’s feedback! Our finetuning strategy follows standard practice in the literature (e.g., Rad-DINO, DINOv2). With the exception of standard DINOv2, all other models already incorporate biomedical data, including CXRs. However, to ensure a fair comparison, we keep the encoders frozen to evaluate how well their foundational features transfer, rather than adapting them to each specific task. Fully finetuning every model would not only be computationally expensive but would also shift the focus from assessing pretrained representations to task-specific adaptation, which isn’t our primary goal.
> > >
> > > For learning rate selection, we completely agree that it plays a significant role in final performance. That’s why we conduct a hyperparameter search for linear probing to ensure optimal settings. However, for full finetuning, we follow the common practice of using predefined learning rates rather than running an exhaustive search, as our focus remains on evaluating pretrained representations rather than optimizing task-specific performance.

---

### Official Review · Reviewer_sLi1 · 2025-02-21

**Confidence:** 4
**Preliminary Rating:** 4
**Recommendation:** Poster
**Final Rating:** 5

**Summary:**

This paper introduces Scan42, a self-supervised vision foundation model tailored for chest X-rays (CXRs), built by adapting the DINOv2 framework. the main contribution of authors is a systematic analysis and adaptation of DINOv2 tailored for medical imaging datasets. They propose 3 key modifications to the ViT architecture:

1- Integrating registers into ViTs to reduce artifacts
2- Replacing Sinkhorn-Knopp centering with moving average centering
3- Reducing prototype heads from 131k to 16k

These changes result in a simplified, faster pipeline while maintaining state-of-the-art accuracy in most of the downstream tasks. Scan42 achieves competitive performance on classification (CheXpert AUROC: 86.80% for ViT-B), segmentation (Dice: 90.94), and radiology report generation (ROUGE-L: 25.25), with 7× fewer FLOPs and 4× less memory than prior work (RAD-DINO). Also, the authors have claimed they will publish the code and weights, helping the reproducibility of the results and open research.

**Strengths:**

The paper’s technical claims are supported by extensive empirical evaluations and ablation studies. The experiments are methodologically sound, with clear quantitative comparisons across multiple benchmarks.

The paper systematically evaluates the impact of individual training components (registers, centering strategies, prototype head dimensions) through well-designed ablation studies.

Rigorous evaluation across classification, segmentation, and report generation using diverse datasets.

The proposed model works really well, demonstrating that image-only pretraining outperforms multimodal methods (e.g., BiomedCLIP) in medical domains, reducing dependency on image-text pairs which are scarce in the medical domain.

**Weaknesses:**

Overall, there is not much significant problem with this paper, however we can mentioned a few:

1- No analysis of why reducing prototypes improves performance or how registers mitigate artifacts.
2- Some sections of the experimental description, especially the ablation studies and certain hyperparameter details, could be presented more clearly to aid reproducibility and reader comprehension.
2- Minor Typographical and Grammatical Issues:
Page 3: "experimental analysis utlizing" → "utilizing."
Page 4: "pretrain compute" → "pretraining compute."
page 4: "comoputation" → "computation."
page 4: "Sare present" → "are presented."
Page 6: "ViT-S Default" → "ViT-S (Default)."
Page 7: "Scan42(S) achieved best in all the metrics" → "Scan42(S) achieves the best performance."
Page 15: "trainable 2-layer MLP projection layer to align visual tokens with textual tokens" → missing hyphen ("two-layer").
page 15: "varient" → "variant."

**Detailed Comments:**

Following are very minor suggestions that could be added to main paper or the corollary.

1- Include definitions or brief explanations of evaluation metrics (e.g., AUROC, AUPRC, Dice score) either in the main text or supplementary material for clarity.

2- Provide additional explanation on the role and impact of "registers" in the Vision Transformer architecture, ideally using a schematic illustration if space permits.

3- Figure 1. could have been prettier with more details.

4- Standardize abbreviations and notations (e.g., ViT, DINOV2, Scan42) consistently throughout the paper

**Justification Of The Final Rating:**

I thank the authors for thoroughly addressing my questions. Although the improvement in model performance remains modest, as other reviewers also pointed out, the paper effectively highlights significant benefits in reducing memory usage and computational complexity (FLOPs). These aspects are particularly valuable to the MIDL audience. Therefore, I have increased my final rating to Strong Accept.

**Justification Of The Preliminary Rating:**

The paper presents a comprehensive empirical analysis with well-designed ablation studies that clearly demonstrate the benefits of the proposed training modifications—most notably the use of registers, moving average centering, and reduced prototype head dimensions. These modifications yield significant efficiency improvements in both compute and model size while achieving competitive or superior performance across multiple downstream tasks. This represents a valuable contribution to the field of medical imaging, addressing the high resource demands typically associated with foundation models.

I am willing to further increase the score to "strong accept", once the authors fix the minor problems and improve the paper with a more comprehensive discussion for the methodology of scan42 and enhanced details on experimental setups (refer to weaknesses).

**Questions To Address In The Rebuttal:**

see "weaknesses" and "Detailed Comments."

**Special Issue:**

No

---

> ### Author Response · Authors · 2025-03-07
> **Rebuttal response to Reviewer sLi1**
>
> - **W1** “No analysis of why reducing prototypes improves performance or how registers mitigate artifacts ”?\
> We thank the reviewer for raising this point. We hypothesize that the improved performance observed with fewer prototypes is due to the implicit regularization effect of reducing parameters, which helps prevent overfitting. Additionally, registers have been empirically shown to enhance performance in transformers by mitigating artifacts. We have updated the manuscript to include this clarification.
> \
> &nbsp;
> Furthermore, biomedical datasets (like CXRs) inherently exhibit a fixed anatomical structure, which reduces the variability and heterogeneity commonly found in natural images. In medical imaging, subtle deviations within this fixed structure are critical for detecting pathological changes. We hypothesize that this characteristic allows us to reduce the number of prototype heads significantly while maintaining downstream task performance, as the model can still effectively capture meaningful variations in CXRs without requiring excessive parameters.
> \
> &nbsp;
> - **W2.1** “Some sections of the experimental description......reproducibility and reader comprehension”?\
> We thank the reviewer for bringing this point. We address the experimental details and request the reviewer to see the revised manuscript and to further enhance reproducibility, we will also release the codebase (upon publication) on github with hyperparameters (described in config.yaml files).
> \
> &nbsp;
> - **W2.2** “Minor typographical and grammatical issues (utlizing->utilizing; comoputation->computation.....;varient->variant)”?\
> We thank the reviewer for their suggestions, and we fixed them in the revised manuscript.
> &nbsp;
> -  **DC1** “Include ........ evaluation metrics (e.g., AUROC, AUPRC).......suppl. material for clarity”?\
> We thank the reviewer for pointing this out and kindly request them to see the revised manuscript as we have clarified to have edited the details in the revised manuscript.
> \
> &nbsp;
> - **DC2** “Provide additional explanation on the role and impact of "registers" ...... illustration if space permits”?\
> We improved Figure 1 to showcase the high-level functionality of registers and talk about their functionality in Section 2.2.
> \
> &nbsp;
> - **DC3** ” Figure 1. could have been prettier with more details.”?\
> We agree with the reviewer, and enhanced Figure 1. We request the reviewer to see it in the revised manuscript.
> \
> &nbsp;
> - **DC4** ” Standardize abbreviations ..... consistently throughout the paper”?\
> We thank the reviewer for helping us improve the readability and consistent abbreviations, and request reviewer to see the revised manuscript.

---

### Official Review · Reviewer_8hJD · 2025-02-25

**Confidence:** 4
**Preliminary Rating:** 3
**Recommendation:** Poster
**Final Rating:** 4

**Summary:**

This paper describes the development of a vision foundation model based on vision transformers (ViT-B, ViT-S) trained using the DINOv2 framework for the analysis of CXRs. The authors make several adaptations to the training of the DINOv2 framework, including adding registers to the DINOv2 checkpoints, centering teacher outputs using a moving average, and reducing prototype heads to 16K for improved efficiency and implicit regularization. The authors conduct experiments using multiple publicly available CXR datasets, and compare performance to other SOTA methods, including RAD-DINO. The experiments primarily show comparable performance to RAD-DINO. The main benefit of the Scan42 models seems to come from the reduced compute needed for pretraining, and the knowledge that these techniques may be used for building other vision foundation models as well. However, the paper does not contain evidence that these optimizations would also translate to other vision foundation model domains.

**Strengths:**

- Well written paper and timely topic given the current interest in VLMs, LLMs, and foundation models.
- Contains ablation experiments which show the improvements obtained.
- The paper uses large publicly available datasets.
- Good comparison to other recent work.
- Models are released on HuggingFace.

**Weaknesses:**

- I am not fully convinced that these improvements would also lead to improvements for other vision foundation models tasks, and the paper does not give evidence for this. Suppose a CT foundation model would be trained using the DINOv2 framework, I am not convinced the presented optimizations would also work well, and the paper does not give evidence that it will. Therefore, the novelty of this paper is potentially limited.

- No GitHub code to reproduce the training experiments. Without this, the added value of this paper is much lower.

**Detailed Comments:**

- How reproducible are the results in Fig 2.b? Could the authors perform 10 runs and plot average and std?

- Table 4: How are the confidence scores obtained? This is from the 500 bootstrapped samples? It would be interesting to see the deviation in performance from different training runs, see previous comment.

- Abstract: Contains the word RSNA. RSNA is the Radiological Society of North America, this is not a dataset. Please rephrase to RSNA Pneunomia dataset throughout the manuscript.

- Other: There are still typos in the manuscript, for example "comoputation". Please do another detailed pass over the manuscript to fix language and typos.

- Other: The authors released their models on HuggingFace, but it would also really help the community if the training code was released on GitHub. Is this possible?

**Justification Of The Final Rating:**

I am satisfied with most of the responses of the authors - they have addressed most of my questions and comments very well! Therefore, I decided to upgrade my rating to Weak Accept. I look forward to see whether the provided optimizations will work well across other vision foundation models in the future! Open-sourcing the codebase will definitely help with that!

**Justification Of The Preliminary Rating:**

I am not fully convinced of (1) the novelty of the work compared to RAD-DINO, (2) whether these optimizations would also work well when building other vision foundation models, and (3) since the code is not released, this is hard for others in the community to build upon.

**Questions To Address In The Rebuttal:**

- Please address my comment whether the authors can make apparent that the suggested optimizations would work well in other domains as well, or are they just task-specific improvements that only work for building a CXR vision foundation model?
- Please consider to add github code.

**Special Issue:**

No

---

> ### Author Response · Authors · 2025-03-07
> **Rebuttal response to reviewer 8hJD**
>
> - **W1** “I am not fully convinced that these improvements would also lead to improvements for other vision foundation models tasks ...”
> We appreciate the reviewer’s insightful comment. We acknowledge that our claims regarding the optimization of the DINOv2 training pipeline are currently specific to chest X-ray (CXR) datasets and do not necessarily extend to other biomedical imaging modalities.
> Medical data availability is often a limiting factor in deep learning research. However, CXRs are an exception, as multiple large-scale publicly available datasets have facilitated the development of foundation models specific to this modality (like RAD-DINO, CheXzero, etc.). To the best of our knowledge, such publicly available large-scale datasets do not yet exist for most other biomedical imaging modalities, which presents a significant challenge in extending our approach beyond CXRs.
> \
> &nbsp;
> That said, we are confident that our optimizations might translate well to other biomedical datasets due to the fixed nature of human anatomy, which reduces the variability and heterogeneity typically present in natural images. In medical imaging, subtle deviations within this fixed anatomical structure are critical for detecting pathological changes—similar to CXRs, where our optimizations have shown success. This underlying consistency across medical modalities suggests that our approach may generalize well, though further validation is needed.
> \
> &nbsp;
> As the reviewer suggests, we are actively exploring the possibility of expanding our study to additional biomedical datasets from diverse sources. However, this is beyond the scope of our current work and will be addressed in future studies. We have revised the discussion section of the manuscript to explicitly clarify this limitation and our plans for future work.
> \
> &nbsp;
> - **W2** “No GitHub code to reproduce the training experiments. ...”?\
> To enhance reproducibility and trust in our results, we agree that we will release the GitHub codebase upon publication, along with the hyperparameters in config.yaml files. We thank the reviewer for their valuable suggestion.
> \
> &nbsp;
> - **DC1** “How reproducible are the results in Fig 2.b? ...”\
> We sincerely appreciate the reviewer's suggestion to perform 10 runs and report the average and standard deviation. Given the high computational cost of pretraining and finetuning, this was unfortunately not feasible within the rebuttal period. However, we want to assure the reviewer that we have taken steps to address variability. Specifically, we follow DINOv2’s evaluation protocol for linear probing, where we experiment with different hyperparameters such as the choice of layers (from the last to 1–4), varying learning rates, and the inclusion of average pool token concatenation. In this setup, we select the classifier that performs best on the validation set and report its performance on the test set. We have updated the manuscript to clarify these details. We recognize the importance of reproducibility and are committed to supporting the community in this regard. To facilitate this, we plan to release our training code along with exact configuration files (.yaml) specifying all hyperparameters, ensuring that our results can be replicated. We hope this clarification addresses the reviewer’s concern, and we appreciate their thoughtful feedback.
> \
> &nbsp;
>
> - **DC2** “How are confidence scores evaluated?..."\
> This is from the 500 bootstrapped samples?...” Yes, the confidence scores are computed using 500 bootstrapped samples, similar to RAD-DINO [1] for LLM-based metrics. This was done to save time and resources needed for pretraining. To ensure consistency across tasks, we use the same bootstrapping strategy for semantic segmentation, report generation, and image classification—reporting performance from 500 bootstrapped samples rather than introducing different evaluation schemes.
> \
> &nbsp;
> - **DC3** “Abstract: Contains the word RSNA........ rephrase to RSNA Pneumonia dataset through the manuscript.”
> \
> We have fixed this in the revised manuscript. Thanks for pointing this out.
> \
> &nbsp;
> - **DC4**” Other: There are still typos in the manuscript....... to fix language and typos.”
> \
> We have fixed this in the revised manuscript. Thanks for pointing this out.
> \
> &nbsp;
> - **DC5** ”Other: The authors released their models on HuggingFace,...... code was released on GitHub. Is this possible?”\
>  We thank the reviewer for the suggestion. To enhance reproducibility and increase trust in our work, we have decided to release our full codebase on Github on acceptance. Unlike RAD-DINO, this release will help the community and contribute to closing the research gap by open-sourcing our work.
>
> References:
> \
> &nbsp;
> [1]: Pérez-García, F., Sharma, H., Bond-Taylor, S. et al. Exploring scalable medical image encoders beyond text supervision, Nature Machine Intelligence,2025

---

### Author Rebuttal · Authors · 2025-03-07

**Rebuttal:**

We appreciate the reviewers' valuable feedback and have carefully considered all the points while revising the manuscript. We kindly request the reviewers to review the revisions and share any further suggestions for improvement.

**Supporting Material:**

/attachment/f33829e70241c2857c3eb0682dbaae4ea850fa1f.pdf

---

### Comment · Area_Chair_kWa2 · 2025-03-08
**Time for discussion and review of the rebuttal**

Dear reviewers

It is now time to consider the responses from the authors. If you are or are not satisfied with author's reply please still post to openreview your feedback to the rebuttal and update your scores. Especially, please update the scores if you feel that the authors have addressed your concerns.

Please note that you can and **are encouraged** to discuss the scores of other reviewers if you disagree with them to make the best

As AC, my responsibility is to post meta-reviews by March 21st, and I would thus like to kindly ask you to consider the authors' rebuttal as soon as possible.

// Your Area Chair

---

### Meta-Review · Area_Chair_kWa2 · 2025-03-24

**Recommendation:** Accept (Poster)
**Confidence:** 4

**Metareview:**

This paper has weak accept, strong accept, and borderline. I did check the borderline review and the paper, and ruling toward accept.